# Large organized chromatin lysine domains help distinguish primitive from differentiated cell populations

Seyed Ali Madani Tonekaboni[1,2], Benjamin Haibe-Kains[1,2,3,4] & Mathieu Lupien [1,2,4✉]

The human genome is partitioned into a collection of genomic features, inclusive of genes, transposable elements, lamina interacting regions, early replicating control elements and cis-regulatory elements, such as promoters, enhancers, and anchors of chromatin interactions. Uneven distribution of these features within chromosomes gives rise to clusters, such as topologically associating domains (TADs), lamina-associated domains, clusters of cis-regulatory elements or large organized chromatin lysine (K) domains (LOCKs). Here we show that LOCKs from diverse histone modifications discriminate primitive from differentiated cell types. Active LOCKs (H3K4me1, H3K4me3 and H3K27ac) cover a higher fraction of the genome in primitive compared to differentiated cell types while repressive LOCKs (H3K9me3, H3K27me3 and H3K36me3) do not. Active LOCKs in differentiated cells lie proximal to highly expressed genes while active LOCKs in primitive cells tend to be bivalent. Genes proximal to bivalent LOCKs are minimally expressed in primitive cells. Furthermore, bivalent LOCKs populate TAD boundaries and are preferentially bound by regulators of chromatin interactions, including CTCF, RAD21 and ZNF143. Together, our results argue that LOCKs discriminate primitive from differentiated cell populations.

[1] Princess Margaret Cancer Centre, Toronto, ON M5G 1L7, Canada. [2] Department of Medical Biophysics, University of Toronto, Toronto, ON M5G 1L7, Canada. [3] Department of Computer Science, University of Toronto, Toronto, ON M5T 3A1, Canada. [4] Ontario Institute for Cancer Research, Toronto, ON M5G 1L7, Canada. ✉email: mathieu.lupien@uhnresearch.ca

The diverse phenotypic identities of each cell type found in multicellular organisms are encoded by lineage-specific biochemically active genomic features, such as transcribed genes[1], active transposable elements[2], anchors of chromatin interactions setting distal boundaries for loop extrusions defining the three-dimensional genome[3], DNA-to-lamin points of contact linking discrete genomic regions to the nuclear lamina[4], early replicating control elements[5] and other *cis*-regulatory elements (CREs) such as promoters and enhancers[6–10]. The uneven distribution of these biochemically detectable features across the genome of any individual cell type underlies an aggregation that gives rise to genomic features of higher order. For instance, clusters of DNA-to-lamin points of contact establish lamina-associated domains (LADs)[4] commonly associated with domains of gene repression[11]. Clusters of contact frequencies between distal genomic regions establish topologically associating domains (TADs)[12] related to active versus inactive chromatin compartments and early versus late origins of DNA replication[12,13]. Similarly, clusters of *cis*-regulatory elements (COREs), defined based on chromatin accessibility, associate with lineage-specific transcription factors and form in proximity of highly expressed lineage-specific essential genes[14]. Moreover, clusters of nucleosomes with post-translationally modified histone lysine residues define large organized chromatin lysine (K) domains (LOCKs) associated with inactive domains when consisting of dimethylated lysine 9 on histone 3 (H3K9me2)[15] or H3K27me3[16]. The comprehensive characterization of post-translationally modified histone lysine residues across the genome of diverse cells, including primitive and differentiated cells, allows to test the relationship between LOCKs with lineage-specific functions.

Here, we show how LOCKs from six post-translational modifications to histone tails (H3K4me1, H3K4me3, H3K27ac, H3K9me3, H3K27me3 and H3K36me3) discriminate primitive from differentiated cells, forming bivalent LOCKs at TAD boundaries in primitive cells enriched for regulators of chromatin interactions. Bivalent LOCKs transit to an H3K9me3-only state as they lose H3K27me3 and H3K4me1 in differentiated cells.

## Results

**Genomic coverage of active LOCKs discriminate ESCs from mature phenotypes**. The Roadmap Epigenomics Project released the complete epigenomes (H3K4me1, H3K4me3, H3K27ac, H3K9me3, H3K27me3 and H3K36me3 from ChIP-seq) across 13 primitive cell types, including embryonic stem cells (ESCs) and induced pluripotent stem cells (iPSCs) as well as 9 ES-derived and 77 differentiated cell types from diverse tissue or origin[6]. Expanding previous work comparing ChIP-seq profiles of histone modifications across stem and differentiated cells conducted on individual elements[17], we used the CREAM tool[14] to identify LOCKs across all 99 aforementioned cell types. Overall, LOCKs of active marks including H3K4me1, H3K4me3 and H3K27ac cover a maximum of 297mbp of the human genome within one cell type, while LOCKs of H3K9me3, H3K27me3 and H3K36me3 repressive marks cover at most 138mbp of the human genome within one cell type (Fig. 1A). Comparing between cell types, LOCKs of the H3K4me1, H3K4me3 and H3K27ac active marks cover a larger proportion of the genome in primitive cells, including ESCs and iPSCs, compared to differentiated cells (non-ESCs and -iPSCs and -ES-derived; FDR < 0.05; Wilcoxon signed-rank test; fold change > 3.1) (Fig. 1A, B). In comparison, the genomic coverage of individual elements for these active histone modifications does not discriminate primitive from differentiated cells (Fig. 1B). In contrast, H3K36me3-, H3K27me3- and H3K9me3-derived LOCKs do not show any significant differences in the proportion of the genome covered between primitive

and differentiated cells (FDR > 0.05; Wilcoxon signed-rank test; fold change < 1.1) (Fig. 1A, B).

**LOCKs of active histone marks are predictive of primitive cell identity**. To further assess the specificity of active histone modifications in identifying primitive from differentiated cellular identity, we developed a k nearest neighbor (k-NN) classifier using LOCKs of each mark as features. Starting from the catalogue of LOCKs for each histone modification we assessed the presence/absence of LOCKs from each histone modification within each cell type over this catalogue. This model shows that LOCKs from active histone modifications stratify primitive from differentiated cell types and cluster each sample according to its tissue of origin (average Matthews Correlation Coefficient (MCC) of active marks = 0.85; repressive marks = 0.71) (Fig. 2). These results parallel previously reported stratification of primitive from differentiated cells using individual active compared to repressive elements[6].

**LOCKs of active histone marks map to cell type-specific biological pathway genes**. *Cis*-regulatory elements (CREs) defined by discrete histone modifications are important players in defining cellular identity by setting lineage-specific gene expression profiles[6–10]. We therefore assessed if LOCKs of active versus repressive marks were related to pathways of relevance to ubiquitous or cell-type-specific biological processes. Cell-type-specific pathways showed higher enrichment among genes in proximity of LOCKs of active marks compared to LOCKs of repressive marks across all cell types (Fig. 3). For example, among the enriched pathways associated with H3K4me1 and H3K4me3 LOCKs we found *EMBRYONIC ORGAN MORPHOGENESIS* in stem cells as well as *LEUKOCYTE CELL CELL ADHESION* in hematopoietic cell populations (FDR < 0.05) (Fig. 3). On the other hand, H3K9me3 LOCKs were enriched in proximity to genes involved in ubiquitous biological processes like *GENE SILENCING* across multiple tissue types (FDR < 0.05) (Fig. 3).

**Bivalency of LOCKs of active histone marks in stem cells**. Coexistence of active and repressive histone modifications at the same loci were reported in primitive cells as bivalent chromatin states associated with genes poised for expression or repression upon cellular differentiation[18]. Hence, we assessed if bivalency is also related to LOCKs[19]. Overlapping repressive marks signal with LOCKs from active and repressed chromatin across our collection of cell types revealed that bivalent LOCKs populate primitive cells, mapping in proximity to genes highly expressed, compared to genes in proximity of individual elements, only in differentiated cells, such as GM12878 and K562 as opposed to primitive H1-hESCs (Fig. 4A). We specifically observed the coexistence of the H3K27me3 repressive LOCKs with H3K4me1 and H3K4me3 active LOCKs in primitive cells (FDR < 0.05) (Fig. 4B). Notably, the H3K27me3 signal intensity did not differ within H3K27me3 LOCKs from the primitive H1-hESC versus the mature GM12878 and K562 cell types (Fig. 4B). We finally assessed the functional classification of genes proximal to bivalent versus active LOCKs found in the H1-hESC primitive cell type through Gene Set Enrichment Analysis. This identified an enrichment of genes proximal to bivalent LOCKs with pathways relevant to embryonic development and stem cell differentiation (FDR < 0.05) (Fig. 4C). Collectively, these results suggest that bivalent LOCKs behave similarly to individual bivalent elements, populating the genome of primitive as opposed to differentiated cell types and being assigned to genes repressed in primitive cells of relevance to differentiation.

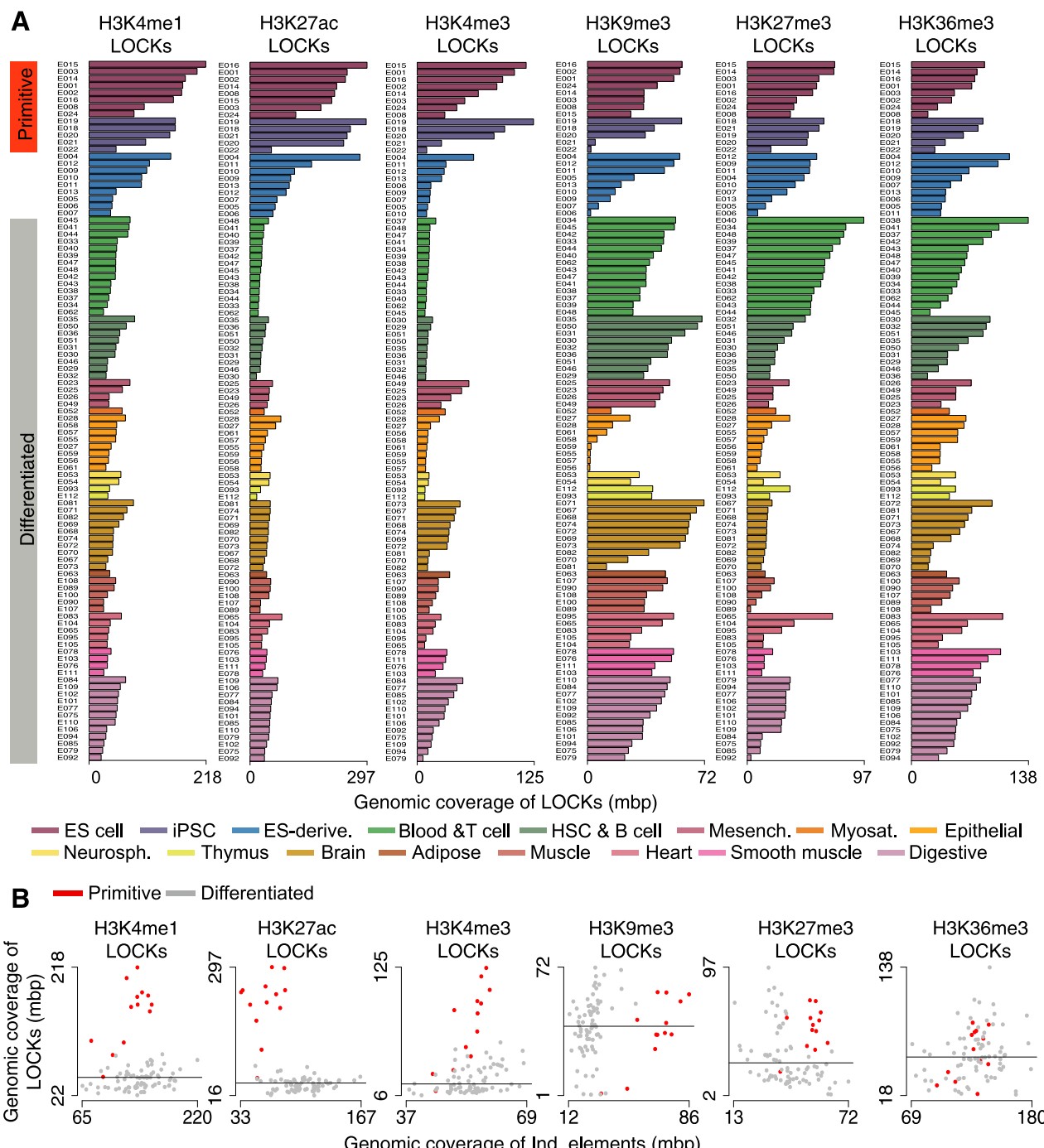

**Fig. 1 Genomic coverage of LOCKs discriminates primitive from differentiated cell types. A** Genomic coverage of LOCKs identified using H3K4me1, H3K4me3, H3K27ac, H3K9me3, H3K27me3 and H3K36me3 histone modification profiles across 13 primitive, 9 ES-derived and 77 differentiated cell types. **B** Comparison of genomic coverage by LOCKs and individual regions (Ind. elements) post-translationally modified with histone marks in 9 primitive and 77 differentiated cell types (ES-derived excluded). Each dot corresponds to one cell type investigated.

**Bivalent LOCKs populate boundaries of TADs**. In addition to LOCKs, the genome is organized into clusters of chromatin interactions defining its three-dimensional organization[20,21]. Clusters of chromatin interactions establish TADs[12] that further cluster into A or B compartments according to the active versus repressed nature of the chromatin within them[13,22]. The three-dimensional genome organization is regulated by DNA binding proteins, namely CTCF, YY1 and ZNF143[23–25] as well as the cohesin complex[20,21]. Previous reports demonstrated the propensity of clusters of CREs and homotypic clusters of

transcription factor binding regions (HCTs) to map at TAD boundaries[14,26]. We therefore investigated the relation between LOCKs, independently from individual elements, and the three-dimensional genome organization of primitive versus differentiated cells. Focusing on H1-hESC, GM12878 and K562, we found that H3K4me1 LOCKs in H1-hESCs were enriched in proximity of TAD boundaries, while the H3K4me1 LOCKs from GM12878 and K562 did not enrich at TAD boundaries (Fig. 5A). This preferentially related to H3K4me1 LOCKs with strong H3K27me3 signal, i.e. bivalent LOCKs (Fig. 5B). H3K4me3 and

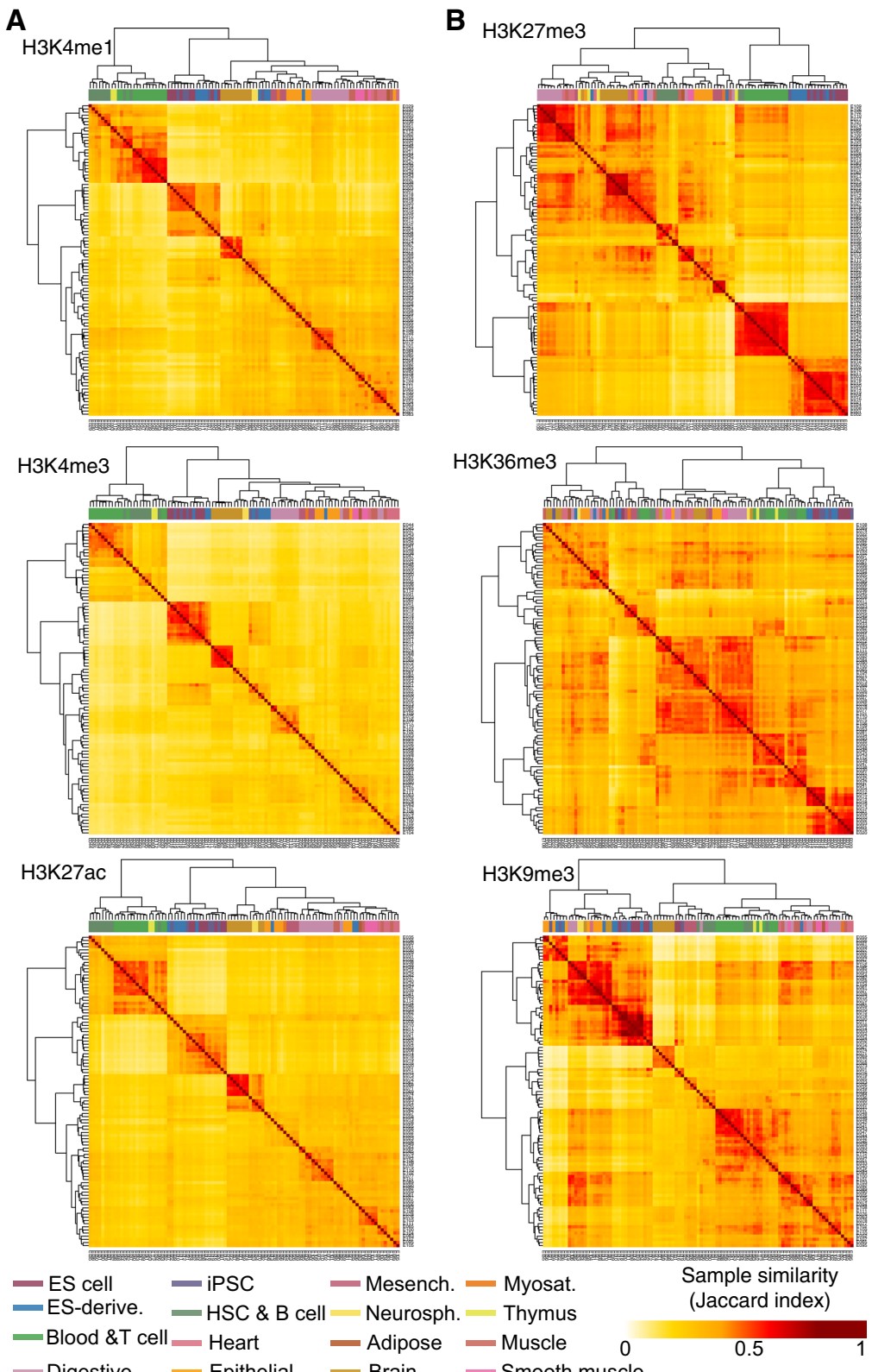

**Fig. 2 LOCKs are predictive of cell identity according to tissue of origin.** Unsupervised clustering of 13 primitive, 9 ES-derived and 77 differentiated cell types according to the similarity in the genomic localization of LOCKs from **A** H3K4me1, H3K4me3, H3K27ac and **B** H3K9me3, H3K27me3, H3K36me3 histone modifications.

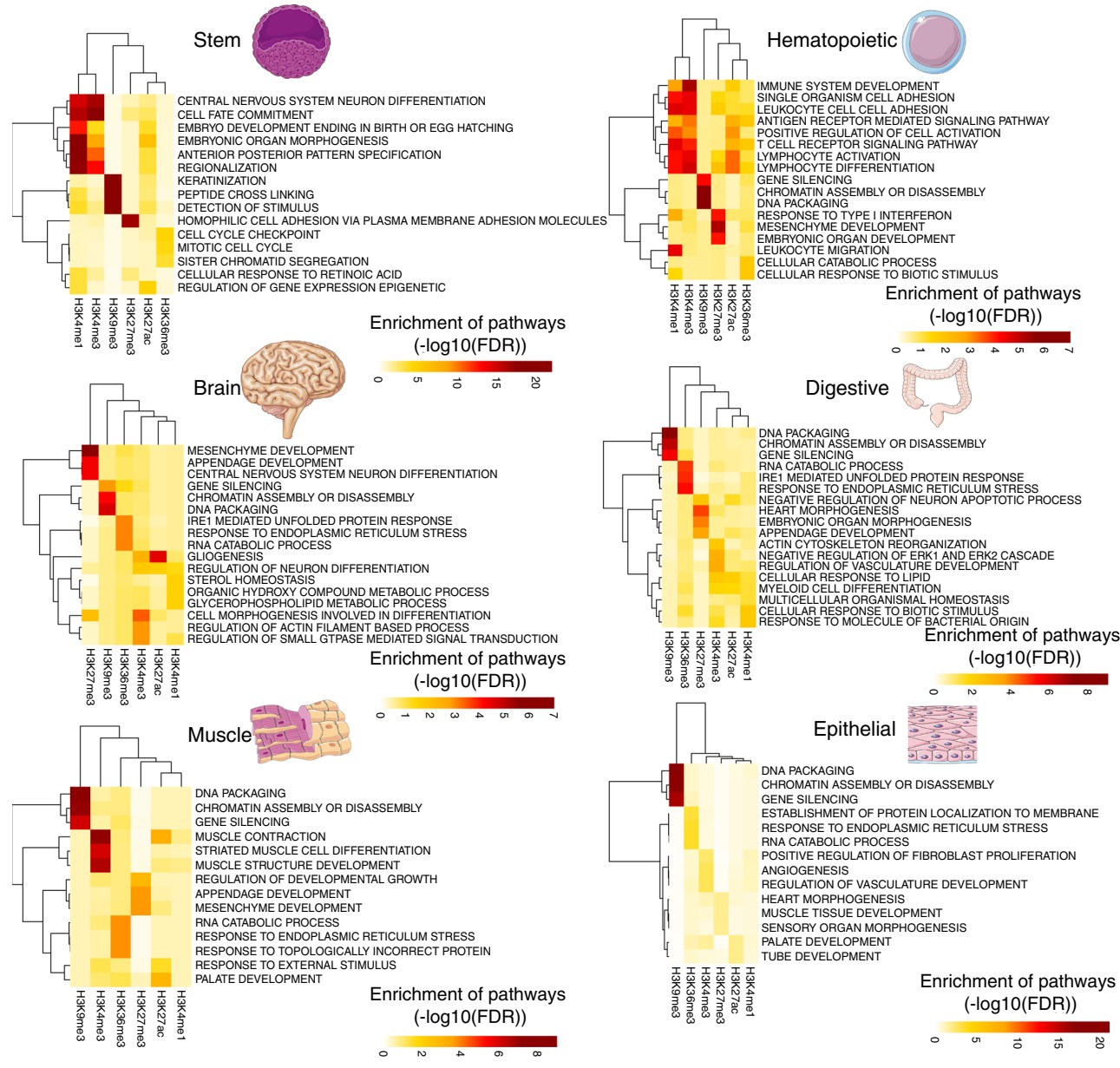

**Fig. 3 Genes associated with LOCKs enrich for pathways related to the tissue of origin.** Gene set enrichment analysis (GSEA) on the collection of genes found within LOCKs of various histone modifications reveals pathways of relevance to stem, hematopoiesis, brain, digestive, muscle and epithelial tissues. In this analysis, the catalogue of genes associated with all the individual elements were used as background gene lists.

H3K27ac LOCKs from primitive and differentiated cell types did not relate to TAD structures (Fig. 5A). We further characterized the H3K4me1 LOCKs with regards to the chromatin occupancy by regulators of chromatin interactions, namely CTCF, YY1, ZNF143 and the cohesin complex component RAD21. This revealed an enrichment of all regulators of chromatin interaction except YY1 at H3K4me1/H3K27me3 bivalent LOCKs from H1-hESC but not over H3K4me1 LOCKs from GM12878 and K562 cell lines (Fig. 5B), exemplified at the chromosome 16q22.1 locus (Fig. 5C). We further showed that the bivalent LOCKs in primitive cells transit to a repressed state in differentiated cells characterized by the gain of the H3K9me3 repressive mark and the loss of H3K4me1 and H3K27me3 (Fig. 5D). Altogether, these results suggest that changes in LOCKs composition discriminating primitive from differentiated cells occur at TAD boundaries.

## Discussion

The human genome is partitioned into diverse genomic features, including transcribed genes, active transposable elements, anchors of chromatin interactions, DNA-to-lamin points of contact, early replicating control elements and other CREs such as promoters and enhancers. These can organize in either individual or clusters[4,12–16]. Here, we studied the transition of clusters of active and repressive histone modifications forming LOCKs across a collection of 13 primitive, 9 ES-derived and 77 differentiated cell types. We show that LOCKs with active marks cover a greater proportion of the genome in primitive cells, such as ESCs, compared to differentiated populations, defining a feature to discriminate the primitive versus differentiated nature of samples. In contrast, LOCKs of repressive histone modifications (H3K9me3, H3K27me3 and H3K36me3) do not discriminate

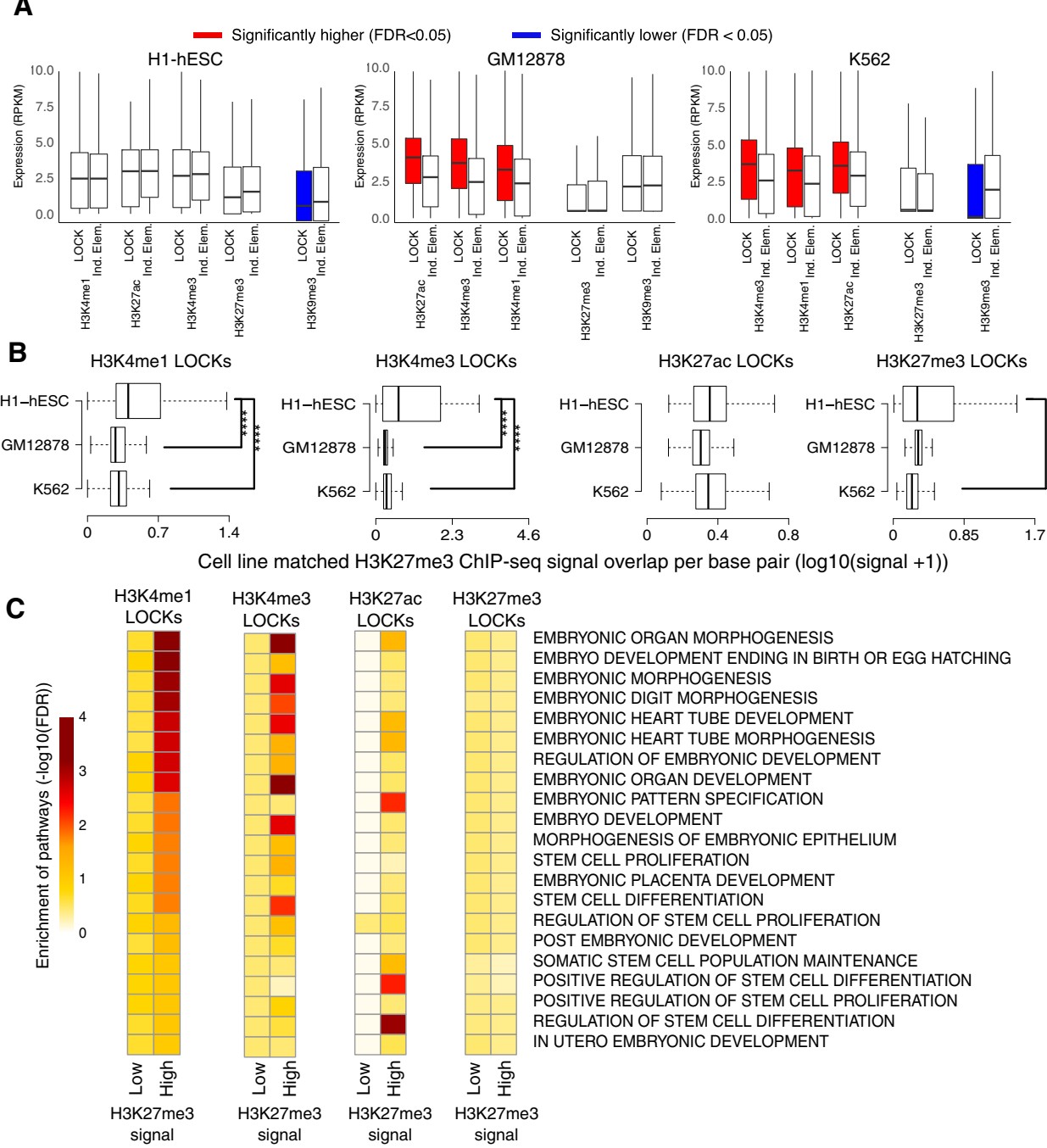

**Fig. 4 Bivalent LOCKs are observed only in primitive cell types and associated with genes repressed in primitive cell types. A** Expression level of genes in proximity of LOCKs versus individual regions (Ind. elements) marked by H3K4me1, H3K4me3, H3K27ac, H3K9me3 or H3K27me3 in the primitive H1-hESC versus differentiated GM12878 and K562 cell types. The boxplots represent the distribution of the expression level of genes (The boxes correspond to the upper and lower quartiles from the median defined by the horizontal line. The whiskers represent the variability outside in gene expression found above or below the quartiles). **B** Quantification of H3K27me3 ChIP-seq signal overlap in LOCKs of active histone modifications (H3K4me1, H3K4me3 and H3K27ac) for H1-hESC, GM12878 and K562 cell types. The signal is normalized (divided by median) to the H3K27me3 ChIP-seq signal overlapped in the Ind. elements of the corresponding profiles in each cell line. Wilcoxon signed-rank test was used to compare distribution of signals in different cell lines (*FDR < 0.05; ****FDR < 0.0001). The boxplots depict the distribution of overlapped ChIP-seq signal bound to be greater than zero and median to be less than 1 (The boxes correspond to the upper and lower quartiles from the median defined by the horizontal line. The whiskers represent the variability outside the lower and upper quartiles). **C** Gene set enrichment analysis (GSEA) reporting the enrichment of pathways in active LOCKs (H3K4me1, H3K4me3 and H3K27ac) and LOCKs of H3K27me3 repressive mark associated or not with elevated H3K27me3 signal in the H1-hESC primitive cell type. In this analysis, the catalogue of genes associated with all the individual elements, with corresponding H3K27me3 signal level, were used as background gene lists.

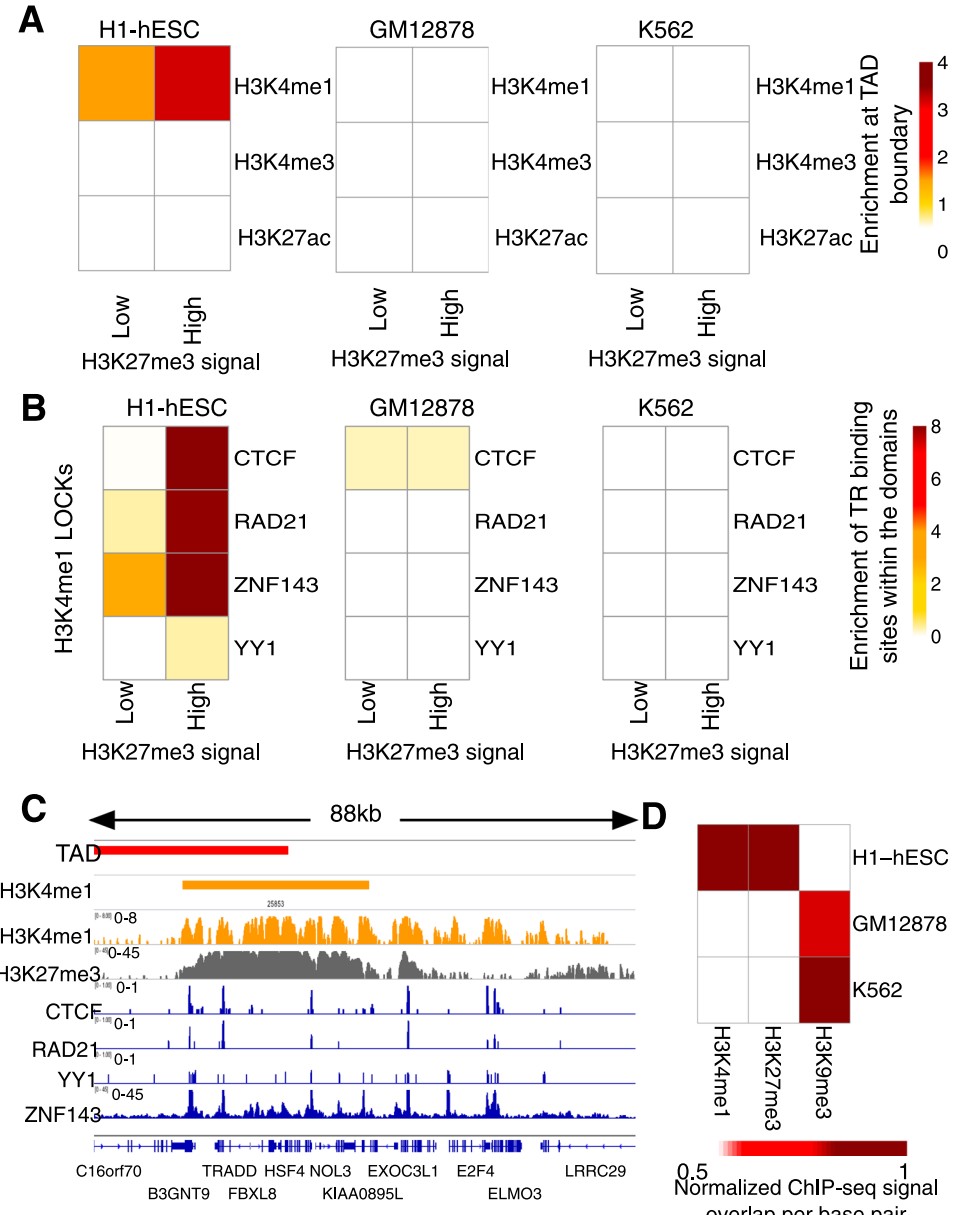

**Fig. 5 Bivalent LOCKs in primitive cells are enriched at TAD boundaries and bound by regulators of chromatin interactions. A** Enrichment of LOCKs of active marks with low or high H3K27me3 ChIP-seq signal at TAD boundaries defined within H1-hESC, GM12878 or K562 cell types (−log10(FDR)). **B** Enrichment of regulators of chromatin interactions (CTCF, RAD21, ZNF143 and YY1) in H3K4me1 LOCKs with low or high H3K27me3 ChIP-seq signal in H1-hESC, GM12878 and K562 cell types (−log10(FDR)). **C** Case example of a bivalent LOCK at a TAD boundary in H1-hESC cells on the chromosome 16q22.1 locus. The ChIP-seq signal intensities for H3K4me1, H3K27me3 and regulators of chromatin interactions are shown. **D** Comparison of the median of H3K4me1, H3K27me3 and H3K9me3 ChIP-seq signal overlap of H1-hESC, GM12878 and K562 cell lines on the H3K4me1 bivalent LOCKs (with high H3K27me3 signal overlap) in H1-hESC. The signal is normalized to the depth of ChIP-seq profiles and size of the LOCKs. In the heatmap, every value for each ChIP-seq profile is also divided by the maximum value of that profile across the cell lines.

primitive from differentiated cells. Collectively, these results align with the permissive as opposed to the restrictive nature of the chromatin reported based on chromatin accessibility in primitive versus differentiated cell types[19].

Although LOCKs with H3K27me3 do not discriminate primitive from differentiated cell types, the co-occurrence of H3K27me3 LOCKs over H3K4me1 LOCKs reflects a bivalent broad chromatin state observed in primitive cell types. Globally, bivalent LOCKs are enriched in proximity to genes involved in stem cell development and differentiation pathways. These results parallel previous reports of bivalent chromatin states at individual regions in primitive versus differentiated cell types

and their relationship with gene expression[18]. Bivalent LOCKs in primitive cell types composed of H3K4me1 and H3K27me3 preferentially map to TAD boundaries and are bound by regulators of chromatin interactions, including CTCF, RAD21 and ZNF143. We also show that these bivalent LOCKs are replaced with H3K9me3 repressive LOCKs in differentiated cells, arguing for a differential role of LOCKs toward the chromatin structure in primitive versus differentiated cells. Altogether, our results demonstrate the utility of studying LOCKs to deepen our understanding of the chromatin-based features discriminating primitive from differentiated populations of cells.

## Methods

**LOCK identification**. We used CREAM algorithm[14] to identify LOCKs. CREAM is used to identify LOCKs in six steps: (1) grouping the individual elements in clusters of varying number of individual elements (referred to as Order); (2) identifying the threshold for the stitching distance between individual elements within the clusters of the same Order; (3) identifying the maximum Order of clusters (LOCKs in this case); (4) clustering individual elements as LOCKs starting from the highest Order; and (5) filtering out low-Order LOCKs with a stitching distance close to the corresponding stitching distance threshold of the same Order. (6) Steps 1–5 are repeated till the relative sum, defined as the sum of coverage of LOCKs by individual elements over the sum of total genomic coverage of LOCKs, starts large oscillations (> 5%).

The code for identifying LOCKs is available in https://codeocean.com/capsule/6911149/tree/v1.

Our approach for LOCK identification relies on identifying clusters of individual elements (peaks) identified by MACS[27] as opposed to the ChIP-seq signal files. This limits the reported challenges in identifying broad domains for certain histone modifications from ChIP-seq signal profiles[28–30].

**Machine learning model for cell type classification**. Similarities between two samples were identified using Jaccard index for the commonality in localization of their identified LOCKs throughout the genome. Then this Jaccard index is used as the similarity statistics in a 1-nearest-neighbor classification approach. The performance of the classification was assessed using the leave-one-out cross-validation.

**Association with genes**. A gene is considered associated with a LOCK or individual element marked by a histone modification if found within 10 kb from each other, with an anchor on the transcription start site (TSS) for genes. This distance was chosen to avoid false-positive association of elements with gene TSSs[31].

**Gene expression comparison**. RNA sequencing profiles of GM12878, K562 and H1-hESC cells lines, available in The ENCODE Project database[32], were used to identify expression of genes in proximity of LOCKs and individual elements marked by a histone modification. Expression levels of genes were compared using the Wilcoxon signed-rank test.

**Pathway enrichment analysis**. Hypergeometric test was used to identify *P*-values for enrichment of gene sets using dhyper function in stats R package (version 3.5.1). LOCK-associated genes in each sample are considered as query gene sets. In case of pathway enrichment per tissue type, a catalogue of genes associated with all the individual elements was used as the background gene list. This catalogue ensures that the identified pathways are specific to LOCKs compared to individual elements.

For the pathway enrichment analysis across LOCKs with different H3K27me3 signal intensity in H1-hESC, all genes in proximity of LOCKs of the same histone mark in H1-hESC are considered as background gene lists. The considered background gene list ensures the specificity of enriched stem cell-related pathways to high H3K27me3 signal LOCKs compared to all LOCKs for a given histone modification.

**Assigning LOCKs to each phenotype**. A LOCK is assigned to each tissue type if it exists in more than 50% of samples from that tissue type.

**H3K27me3 signal intensity measurement over active LOCKs**. We identified an overlap of the signal from bedgraph files of ChIP-Seq data of H3K27me3 with the identified LOCKs and individual elements in GM12878, K562 and H1-hESC using bedtools (version 2.23.0). The signal over LOCKs was normalized (divided by median) to the H3K27me3 ChIP-seq signal overlapped in the Ind. elements of the corresponding profiles in each cell line. The identified signal intensity within each LOCK was then further normalized to the length of the element. The distributions of the normalized scores were then compared between H1-hESC and GM12878 and K562 cell lines using Wilcoxon signed-rank test.

**Enrichment of LOCKs at boundaries of TADs**. TAD boundaries identified from a collection of Hi-C profiles from GM12878, K562 and H1-hESC that were processed for genome assembly GRCh37 are available in the Hi-C browser[33]. LOCKs identified from ChIP-seq profiles of histone modification in a cell line were categorized to be at TAD boundaries if within 10 kb from each other. A hypergeometric test was then used to identify enrichment of LOCKs from a histone modification of a cell line with a defined H3K27me3 overlap level, low or high. The LOCKs in each category of high, low or intermediate H3K27me3 signal overlap are considered as the query set and all the LOCKs at TAD boundaries as the background. Considering all LOCKs as the background list ensures the specificity of the enrichment at TAD boundaries to high H3K27me3 signal LOCKs compared to all LOCKs of a given histone modification.

**Enrichment of binding sites for regulators of chromatin interactions at LOCKs**. The number of individual binding sites for regulators of chromatin interactions within H3K4me1 LOCKs associated with low or high H3K27me3 signal from H1-hESC cells was normalized to the size of the LOCKs. The normalized binding value was then used as a query set in a hypergeometric test to measure their enrichment score. The normalized binding value of regulators of chromatin interactions over all H3K4me1 LOCKs in H1-hESC is considered as the background list within the hypergeometric test.

**Multiple hypothesis correction**. *P*-values were corrected for multiple hypothesis testing using the Benjamini–Hochberg procedure[34].

**Reporting Summary**. Further information on research design is available in the Nature Research Reporting Summary linked to this article.

## Data availability

All the histone modification ChIP-seq data used in this manuscript have been publicly available through ENCODE and Roadmap project Consortiums and can be accessed at "ChIP-seq [https://www.encodeproject.org/search/?type=Experiment&related_series.%40type=ReferenceEpigenome&replicates.library.biosample.donor.organism.scientific_name=Homo+sapiens&replicates.library.biosample.life_stage=adult&replicates.library.biosample.life_stage=embryonic&award.project=Roadmap&assay_title=Histone+ChIP-seq]". In addition, for transparency and reproducibility purposes, we provide a copy of all the call peak files used to generate figures of the manuscript in the Supplementary Software. We also provide the data and our analytical pipeline in the cloud-based computational reproducibility platform CodeOcean (https://codeocean.com/capsule/6911149/tree/v1). All other relevant data supporting the key findings of this study are available within the article and its Supplementary Information files or from the corresponding author upon reasonable request. A Reporting Summary for this Article is available.

## Code availability

All the codes for generating the results of this manuscript are available in cloud-based computational reproducibility platform CodeOcean (https://codeocean.com/capsule/6911149/tree/v1).

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

## Acknowledgements

We acknowledge the Roadmap Epigenomic, the ENCODE Consortium, the ENCODE production laboratories and the ENCODE Data Coordination Center that generated the data sets used in this manuscript. This study was conducted with the support of the Terry Fox Research Institute (New Frontiers Research Program PPG-1064, M.L. and B.H.K.), Canadian Cancer Research Society and the Ontario Institute for Cancer Research through funding provided by the Government of Ontario. This work was also supported by the Canadian Institute for Health Research (CIHR: Funding Reference Number 136963 to M.L. and 363288 to B.H.K.) and Princess Margaret Cancer Foundation (M.L. and B.H.K.). We acknowledge the Princess Margaret Bioinformatics group for providing the infrastructure assisting us with the analysis presented here. M.L. holds an Investigator Award from the Ontario Institute for Cancer Research, the Bernard and Francine Dorval Award of Excellence from CCSRI. S.A.M.T. was supported by Connaught International Scholarships for Doctoral Students. B.H.K. is supported by the Gattuso-Slaight Personalized Cancer Medicine Fund at Princess Margaret Cancer Centre and the Canadian Institutes of Health Research.

## Author contributions

S.A.M.T. developed and performed the analysis. S.A.M.T., B.H.-K. and M.L. interpreted the results and conceived the design of the study. S.A.M.T., B.H.-K. and M.L. wrote the manuscript. B.H.-K. and M.L. supervised the study.

## Competing interests

The authors declare no competing interests.
