## [Peer Review File · Nature Communications]

Reviewers' comments:

Reviewer #1 (Remarks to the Author):

In the manuscript "Large Organized Chromatin Lysine Domains (LOCKS) Distinguish Primitive from Differentiated Cell Populations", the authors present analysis results where they apply their previously published CREAM tool (Tonekaboni, et al 2019) to histone modification data from the Roadmap Epigenomics and ENCODE projects.

A major concern I have with this manuscript is the limited novelty with most of the biological results being presented. In some of the cases the authors are not acknowledging relevant prior literature.

Part of the results of Fig. 1 appears closely related to the result of Fig. 2d of Zhu et al, Cell 2013. Fig. 2d of Zhu et al showed that a larger % of the genome is within 50kb of H3K4me1 sites in ESC/iPS than other cell types. The Zhu et al, Cell 2013 paper was not cited despite originally publishing many of the datasets the authors are analyzing in this paper. Also the authors cite the H3K9me2 result of Wen et al, 2009, but there was no reference to the literature discussing the controversy/subtleties related to that result (see Filion and van Steensel, Nature Genetics 2010; Wen et al Nature Genetics 2010; Hawkins et al, Cell Stem Cell 2010; Linert et al, PLoS Genetics 2011; Zhu et al, Cell 2013).

Fig. 2-4 show results that are relatively standard without LOCKs (i.e. showing cell types cluster based on epigenomic marks, different gene set enrichments in the presence of different epigenetic marks, and properties of bivalent regions). It was not clear there was any advantage or new insights by conducting the analysis through LOCKs opposed to in more direct manners.

Fig. 5 shows differences with TADs and active marks in ESC and differentiated cells. There was no effort to relate this result to other papers in the literature that have already gone deeper in showing differences between ESC and differentiated cells in terms of TADs and histone marks (e.g. Bonev et al Cell 2017; Barrington, et al Nature Communications 2019).

Additional comments

1. The author took an existing term LOCKs used in a similar context, but then changed its definition, which will confuse the literature. Wen et al defined "Large Organized Chromatin K9-modifications (LOCKS)" while here the authors refer to LOCKs as any lysine domain and not specifically K9.

2. Last sentence of abstract "Together, our results argue that LOCKs are determinant features defining the identity of primitive cell populations and their transition to differentiated phenotypes" – 'determinant' has a causal interpretation, but only correlations were shown in the manuscript

3. 90 cell types were stated as being used, but should that have been 100 (13 primitive and 87 differentiated)? Also why weren't all 127 epigenomes from the Roadmap Epigenomics et al, 2015 paper used in the analysis?

4. Why for the analysis in Fig. 4 was there a switch from using the large collection of Roadmap Epigenomics data to just three ENCODE cell types? Also note while the analysis of Fig. 5 is limited by cell types with Hi-C data, many more ENCODE/Roadmap Epigenomic cell types have Hi-C data available (e.g. Schmitt et al, Cell Reports 2016).

5. LOCK were discussed as covering specific absolute number of bases. Those values likely depend on arbitrary parameter settings within the CREAM method. How sensitive are the results to those parameters values?

6. On page 4, when reporting that one thing is above a significance threshold and another is below the same threshold, that does not on its own imply a significant difference.
7. 'Average MCC' – not defined what MCC refers to
8. Figure 1, should be rotated
9. Fig 5, the dynamic range of the color scale should be adjusted to better differentiate among values that are currently all white
10. Typo "H3K37me3"
11. I did not see the actual LOCKs calls provided
12. Cited a Goodnight et al, but did not provide a reference
13. Deblois et al, missing details in reference list

Reviewer #2 (Remarks to the Author):

The overarching theme of this manuscript is to ask if patterns of histone modifications (specifically H3K4me1, H3K4me3, H3K27ac, H3K27me3, H3Kme9, and H3K36me3) in Large Organized Chromatin Lysine Domains, or LOCKs, discriminate primitive cell types from differentiated. The authors utilized their CREAM tool to first define LOCKs for each of the six aforementioned histone modifications across 100 cell types (13 primitive and 87 differentiated) from the RoadMap Consortium. They found that active LOCKs have higher genome coverage in primitive cells versus differentiated cells, although there was no significant difference in genome coverage for repressive marks.

Turning the focus next to bivalent domains in primitive cells, the authors found that LOCKs behave similarly to bivalent histone marks on the individual level, with LOCKs harboring both active and repressive marks in primitive cell types. It was also revealed that bivalent LOCKs in these primitive cell types map to TAD boundaries and are succeeded by H3K9me3 repressive marks upon cell differentiation. This is a well-presented manuscript and I have only a few concerns/suggestions outline below.

Minor Concerns

- In the "Genomic Coverage..." section at the beginning of the manuscript, the authors cite a 2015 Roadmap paper, but in methods, they state used both ENCODE and Roadmap ChIP-seq data. Could this be more clearly resolved in the text?
- Figure 1 is very hard to read/interpret. Perhaps cell type names in A are not needed and move the labels to the right down below? The positioning of B, small red/gray dots and the direction of text made it take a while to figure out what the authors are showing.
- In figure 4A, H3K36me3 is mentioned in the caption but not actually plotted anywhere. I assume it's not looked at in the Bivalent analysis and should be removed from the caption.
- On page four and later on throughout the paper, it is noted that the Roadmap Epigenomics Consortium's histone ChIP-seq data from 13 primitive and 87 differentiated cell types were used. The manuscript then proceeds to refer to these as "these 90 cell types," so I assume there are redundant ones in the 100? This should be mentioned in the text.
- Along those lines, the definition of stem vs. differentiated isn't clear in Figures. Are they calling ES-derived differentiated? If so in Fig 1A there is one outlier sample for K4me3 and K27ac that is higher than many ES/iPS cells. What sample is this and is it worth discussing in the text?
- I personally find it interesting that the hematopoietic cells are highest in the repressive LOCKs,

and lowest in the active marks. I recall trying to sonicate B-cell chromatin and it taking three or four times longer than ES or fibroblast chromatin to shear, and wonder if this is one reason why (others have probably described genome compaction in hematopoietic cells)

- A few typos, "marked by an histone modification" "H3K27me3 do not discriminating primitive"

Major Concerns

- In the results section, the authors state that "...H3K36me3, H3K27me3, and H3K9me3 derived LOCKs do not show any significant differences in the percentage of the genome covered between primitive and differentiated cells." They later appear to contradict this statement in the conclusion, though, with the statement that "LOCKs with the H3K9me3 repressive histone modifications span over a greater proportion of the genome in differentiated versus primitive cells." This also contrasts the summary section, as well.
- Figure 1 shows the basis of the paper, that active mods K4me/27ac LOCKs are higher in stem vs. differentiated cells. However, in 1A you can see there is quite a bit of variation across the ES/iPS cells (K4me1/3 especially) so I'm not sure how the FDR of >0.05 was calculated. Is there a better test to show significance? The ES-derived were already mentioned and depending on which group the authors are considering them makes the variation more striking. It should be addressed in the text why some ES/iPS samples are lower than many differentiated cells. This could be a technical artifact of that ChIP-seq having lower enrichment, or something about the biology of that sample.

We thank the reviewers for their thoughtful comments for our manuscript. These have helped us improve on the quality of our work and better showcase the importance of LOCKs and their features discriminating primitive versus differentiated in the context of previous work. Please find below our point by point responses to the reviewers comments.

Reviewer #1:

In the manuscript “Large Organized Chromatin Lysine Domains (LOCKS) Distinguish Primitive from Differentiated Cell Populations”, the authors present analysis results where they apply their previously published CREAM tool (Tonekaboni, et al 2019) to histone modification data from the Roadmap Epigenomics and ENCODE projects.

A major concern I have with this manuscript is the limited novelty with most of the biological results being presented. In some of the cases the authors are not acknowledging relevant prior literature.

We have revised the manuscript to improve the description of our work to focus on the innovative approaches relying on assessing the distinction between clusters of modified histones forming LOCKs across cell types in contrast to the traditional focus on individual regions reported from ChIP-seq peak callers. We also appreciate the suggestions from the reviewer regarding specific additional published work for us to reference in our manuscript.

Part of the results of Fig. 1 appears closely related to the result of Fig. 2d of Zhu et al, Cell 2013. Fig. 2d of Zhu et al showed that a larger % of the genome is within 50kb of H3K4me1 sites in ESC/iPS than other cell types. The Zhu et al, Cell 2013 paper was not cited despite originally publishing many of the datasets the authors are analyzing in this paper.

The analysis conducted in Zhu et al looks at the proportion of the genome capture within 50kb from individual peaks of modified histones (H3K4me1 or others) independently of whether peaks for a given histone modification are clustered or not (i.e. if they form LOCKs or not). In contrast, our approach specifically calculates the proportion of the genome within LOCKs of modified histones (H3K4me1 or others types of LOCKs). Hence, our work differs from the previous publication in that it is focused on LOCKs as opposed to individual peaks. We have modified our manuscript to reflect the difference with the work from Zhu et al on the first paragraph of the results section on page 4 as follows:

“The Roadmap Epigenomics Project released the complete epigenomes (H3K4me1, H3K4me3, H3K27ac, H3K9me3, H3K27me3 and H3K36me3 from ChIP-seq) across 13 primitive cell types, including Embryonic Stem Cells (ESCs) and induced Pluripotent Stem Cells (iPSCs) as well as 9 ES-derived and 77 differentiated cell types from diverse tissue or origin⁶. Expanding previous work comparing ChIP-seq profiles of histone modifications across stem and differentiated cells conducted on individual elements (Zhu et al 2013)”

Also the authors cite the H3K9me2 result of Wen et al, 2009, but there was no reference to the literature discussing the controversy/subtleties related to that result (see Filion and van Steensel, Nature Genetics 2010; Wen et al Nature Genetics 2010; Hawkins et al, Cell Stem Cell 2010; Linert et al, PloS Genetics 2011; Zhu et al, Cell 2013).

With regards to the controversy in detecting broad repressive histone modification domains raised by Filion and van Steensel (Nature Genetics 2010) or the broad peak detection reported by others, it is important to note that our method differs from Wen et al 2009. Our method to call LOCKs relies on the CREAM tool that uses “narrow peaks” as opposed to the ChIP-seq signal file to call clusters of narrow peaks. Indeed, we used narrow peaks provided by The ENCODE Project and the Roadmap Epigenomics Project that were identified using peak callers using the local background to report peaks. Our approach is therefore less incline to being affected by local broad signal distributions in ChIP-seq data. We have clarified this point in the methods sections, in light of the previous work from the field stated in the reviewer’s comment, in page 15 as follows

“Our approach for LOCK identification relies on identifying clusters of individual elements (peaks) identified by MACS (Zhang et al. 2008) as opposed to the ChIP-seq signal files. This limits the reported challenges in identifying broad domains for certain histone modifications from ChIP-seq signal profiles (Filion and van Steensel 2010; Hawkins et al. 2010; Lienert et al. 2011). ”

Fig. 2-4 show results that are relatively standard without LOCKs (i.e. showing cell types cluster based on epigenomic marks, different gene set enrichments in the presence of different epigenetic marks, and properties of bivalent regions). It was not clear there was any advantage or new insights by conducting the analysis through LOCKs opposed to in more direct manners.

Previously published analyses were performed by comparing all elements (narrow or broad) independently of whether they formed LOCKs or not. We expanded those analyses by subdividing elements into those that cluster into LOCKs from those that do not. Comparing these two populations of elements for different histone modifications, we demonstrate that LOCKs readily discriminate primitive from differentiated populations while this is not the case for individual elements. This is presented in its clearest form in Fig 1B where the individual elements are compared to LOCKs for each histone modification across primitive and differentiate cell types. Collectively, this argues that the biology previously reported stemmed from the unidentified LOCKs embedded with the sum of all elements.

The novelty of our work lies in improving the stratification of peaks reported from ChIP-seq of histone modifications to refine our understanding of biological processes. The concept of stratifying elements has been applied before in the field, for instance in assigning elements to promoters versus distal elements. In our case, we dismissed the human biased categorization

of the genome and applied instead a machine learning method relying on the intrinsic nature of the data to discriminate clusters forming LOCKs from individual elements.

Fig. 5 shows differences with TADs and active marks in ESC and differentiated cells. There was no effort to relate this result to other papers in the literature that have already gone deeper in showing differences between ESC and differentiated cells in terms of TADs and histone marks (e.g. Bonev et al Cell 2017; Barrington, et al Nature Communications 2019).

We greatly appreciate the work from Bonev et al 2017 and Barrington et al 2019 and acknowledge their important contribution to our understanding of TADs as they differ between primitive and differentiated mouse cells. Because our work is focused on presenting the features of LOCKs that discriminate primitive from differentiated cells, we did not want to distract the reader towards other concepts. Hence, to address the reviewer's request while maintaining the focus on LOCKs we have revised the manuscript to capture the contributions of others in the 3D genome organization field with the following sentence that includes the reference to the review on the topic by Juanma Vaquerizas, one of the leaders in the field of TADs and development, on page 9.

“In addition to LOCKs, the genome is organized into clusters of chromatin interactions defining its three-dimensional organization (Hug and Vaquerizas 2018).”

Additional comments:

1. The author took an existing term LOCKs used in a similar context, but then changed its definition, which will confuse the literature. Wen et al defined “Large Organized Chromatin K9-modifications (LOCKS)’ while here the authors refer to LOCKs as any lysine domain and not specifically K9.

We acknowledge that the term LOCK was first introduced by Dr. Andrew Feinberg based on H3K9 modifications. We also acknowledge that our field has been overwhelmed with different terms for very similar elements, such as super-enhancers as opposed to COREs, neighbourhoods as opposed to TADs, etc. With the intent to limit the creation of new terms for elements related in concept, we consulted directly with Dr. Feinberg to get his approval to use the term LOCKs for more than just H3K9 modifications (correspondence emails shared with the editor for privacy reasons). Our intention is not to confuse the literature but rather to keep it accessible to a broad audience.

2. Last sentence of abstract “Together, our results argue that LOCKs are determinant features defining the identity of primitive cell populations and their transition to differentiated phenotypes” – ‘determinant’ has a causal interpretation, but only correlations were shown in the manuscript

We have revised the manuscript to:

“Together, our results argue that LOCKs discriminate primitive from differentiated cell populations, as they relate to transcription regulating events.”

3. 90 cell types were stated as being used, but should that have been 100 (13 primitive and 87 differentiated)? Also why weren't all 127 epigenomes from the Roadmap Epigenomics et al, 2015 paper used in the analysis?

We appreciate the reviewer's comment, also raised by reviewer 2. This was a typo that has now been corrected to 99. We used 99 cell types because we restricted our analyses to models assigned to a specific cell/tissue type. We excluded all samples categorized as “IMR90”, “Other” and “ENCODE 2012” in Fig 2 of the Kundaje et al 2015 Nature 518. p7539 publication from our manuscript. However, we identified LOCKs for all datasets from the 127 epigenomes and made them available through our CodeOcean capsule (<https://codeocean.com/capsule/6911149/tree/v1>).

4. Why for the analysis in Fig. 4 was there a switch from using the large collection of Roadmap Epigenomics data to just three ENCODE cell types? Also note while the analysis of Fig. 5 is limited by cell types with Hi-C data, many more ENCODE/Roadmap Epigenomic cell types have Hi-C data available (e.g. Schmitt et al, Cell Reports 2016).

We restricted our analyses in Figure 4 and 5 to the Tier I cells lines from The ENCODE Project that have been deeply characterized and studied and for which RNA-seq, Hi-C and ChIP-seq profiles of histone modifications were available through the ENCODE data portal when we started this work. This was meant to lighten the information content to help the reader understand the results considering that we were combining additional datasets for each cell type.

5. LOCK was discussed as covering a specific absolute number of bases. Those values likely depend on arbitrary parameter settings within the CREAM method. How sensitive are the results to those parameters values?

Details of the CREAM method were publicly released in our 2019 Genome Research publication (doi:10.1101/gr.248658.119). CREAM has one hyperparameter (WScutoff). This parameter is identified automatically to identify LOCKs of each ChIP-seq profile. This automatic identification of WScutoff takes care of differences in the size of peaks and their genomic distribution specific to each sample. We now provide a CodeOcean capsule for the CREAM analysis included in this manuscript <https://codeocean.com/capsule/6911149/tree/v1>.

6. On page 4, when reporting that one thing is above a significance threshold and another is below the same threshold, that does not on its own imply a significant difference.

We have revised our statements on page 4 to clarify how significance was determined.

“Comparing between cell types, LOCKs of the H3K4me1, H3K4me3 and H3K27ac active marks cover a larger proportion of the genome in primitive cells, such as ESCs, compared to differentiated cells (FDR < 0.05; Wilcoxon signed-rank test) (Fig. 1A-B). In comparison, the genomic coverage of individual elements for these active histone modifications does not discriminate primitive from differentiated cells (Fig 1B). In contrast, H3K36me3, H3K27me3 and H3K9me3 derived LOCKs do not show any significant differences in the proportion of the genome covered between primitive and differentiated cells (FDR > 0.05; Wilcoxon signed-rank test) (Fig. 1A-B).”

7. ‘Average MCC’ – not defined what MCC refers to

MCC refers to Matthews Correlation Coefficient. We modified the text accordingly in the revised manuscript

8. Figure 1, should be rotated

Thank you for the suggestion. We have rotated Figure 1 in the revised manuscript.

9. Fig 5, the dynamic range of the color scale should be adjusted to better differentiate among values that are currently all white

We have set the transition from white to color to distinguish significant (color) from non-significant (white) enrichments. We are concerned that changing this range would be misleading.

10. Typo “H3K37me3”

Changed to “H3K27me3”

11. I did not see the actual LOCKs calls provided

The called LOCKs are provided as bed files as supplementary material.

12. Cited a Goodnight et al, but did not provide a reference

The reference has been removed in the revision process.

13. Deblois et al, missing details in reference list

Details are now provided in the revised version of the manuscript.

Reviewer #2 (Remarks to the Author):

Minor Concerns:

1) In the “Genomic Coverage...” section at the beginning of the manuscript, the authors cite a 2015 Roadmap paper, but in methods, they state used both ENCODE and Roadmap ChIP-seq data. Could this be more clearly resolved in the text?

The sentence in the results section is there to acknowledge that the “complete epigenomes” across all the cells/tissue types we used were collectively presented by the Roadmap Epigenomic Project in their Nature 2015 publication. That said, the ChIP-seq data employed in that manuscript were generated as part of The ENCODE Project and the Roadmap Epigenomic Project. Hence, we acknowledge both sources in the method and acknowledgement section. We revised our acknowledgement to reflect the contribution of both projects to the ChIP-seq data. We also now provide a CodeOcean capsule for the CREAM analysis included in this manuscript that ensures others can reproduce our analyses using the files we collected from The ENCODE Project and the Roadmap Epigenomics Project (<https://codeocean.com/capsule/6911149/tree/v1>).

2) Figure 1 is very hard to read/interpret. Perhaps cell type names in A are not needed and move the labels to the right down below? The positioning of B, small red/gray dots and the direction of text made it take a while to figure out what the authors are showing.

The figure 1 has been modified to address its readability according to comments from both reviewers 1 and 2.

3) In figure 4A, H3K36me3 is mentioned in the caption but not actually plotted anywhere. I assume it's not looked at in the Bivalent analysis and should be removed from the caption.

Allusion to H3K36me3 has been removed from Figure 4A caption in the revised version of the manuscript.

4) On page four and later on throughout the paper, it is noted that the Roadmap Epigenomics Consortium's histone ChIP-seq data from 13 primitive and 87 differentiated cell types were used. The manuscript then proceeds to refer to these as “these 90 cell types,” so I assume there are redundant ones in the 100? This should be mentioned in the text.

We actually used data from 99 cell types, namely 13 labelled as primary cultures, 9 ES-derived and 77 differentiated cell types in the Kundaje et al Nature 2015 manuscript (we excluded the “IMR90”, “ENCODE 2012” and “Other” labelled samples). We have replaced “90” with the detailed numbers for each grouping (13, 9 and 77). To help others in the field navigate LOCKS across all samples included in the Kundaje et al 2015 Nature manuscript, we provide LOCKs on

all epigenomes from that publication in our CodeOcean capsule (<https://codeocean.com/capsule/6911149/tree/v1>).

5) Along those lines, the definition of stem vs. differentiated isn't clear in Figures. Are they calling ES-derived differentiated? If so in Fig 1A there is one outlier sample for K4me3 and K27ac that is higher than many ES/iPS cells. What sample is this and is it worth discussing in the text?

Primitive cell types exclude the ES-derived samples. Differentiated cell types also exclude ES-derived samples. ES-derived samples are presented in figure 1A and 2 to allow for their comparison with other samples but our analyses for significant differences between primitive and differentiated cell types are restricted to the comparison of the 13 primitive versus the 77 differentiated cell types. This is now clearly indicated across the manuscript.

In terms of outliers, the sample referred to by the reviewer is E004 - H1 BMP4 derived mesoderm. Our data includes another derived mesoderm (E013 - HUES64 derived CD56+ mesoderm) that does not show the same trends for H3K4me1 and H3K27ac LOCKs. Hence, we cannot conclude that ES derived mesoderms behave differently from any other type of ES derived cells.

6) I personally find it interesting that the hematopoietic cells are highest in the repressive LOCKs, and lowest in the active marks. I recall trying to sonicate B-cell chromatin and it taking three or four times longer than ES or fibroblast chromatin to shear, and wonder if this is one reason why (others have probably described genome compaction in hematopoietic cells)

Global chromatin compaction measured based on DNase-seq (Stergachis et al. 2013) demonstrates that pluripotent/stem cells have more accessible chromatin than mature populations. Similarly, physical plasticity of the nucleus measured by micromanipulation is highest in stem compared to differentiated cells (reviewed in DOI: [10.1016/S0091-679X\(10\)98009-6](https://doi.org/10.1016/S0091-679X(10)98009-6)). These observations align with more repressive LOCKs and less active LOCKs in mature versus pluripotent hematopoietic cells and with differences in sonication conditions between primitive and differentiated cells. However, we are not aware of a systematic assessment of sonication conditions across the 99 cell types included in this study that could serve to adequately investigate the relationship between LOCKs and the ease of chromatin fragmentation.

7) A few typos, "marked by an histone modification" "H3K27me3 do not discriminating primitive"

We have carefully reviewed the manuscript for typos and addressed those mentioned by the reviewer.

Major Concerns

1) In the results section, the authors state that "...H3K36me3, H3K27me3, and H3K9me3 derived LOCKs do not show any significant differences in the percentage of the genome covered between primitive and differentiated cells." They later appear to contradict this statement in the conclusion, though, with the statement that "LOCKs with the H3K9me3 repressive histone modifications span over a greater proportion of the genome in differentiated versus primitive cells." This also contrasts the summary section, as well.

We have corrected the statements in the conclusion and summary to align with the results.

2) Figure 1 shows the basis of the paper, that active mods K4me/27ac LOCKs are higher in stem vs. differentiated cells. However, in 1A you can see there is quite a bit of variation across the ES/iPS cells (K4me1/3 especially) so I'm not sure how the FDR of >0.05 was calculated. Is there a better test to show significance?

We used Wilcoxon signed-rank test to compare the genomic coverage in stem versus differentiated samples for each histone modification. As a non-parametric test, Wilcoxon signed-rank test does not have any assumption regarding the distribution of coverage by histone mods in stem or differentiated samples. We revised the text, in page 4, as follows to specify what test we used.

"Comparing between cell types, LOCKs of the H3K4me1, H3K4me3 and H3K27ac active marks cover a larger proportion of the genome in primitive cells, such as ESCs, compared to differentiated cells (FDR < 0.05; Wilcoxon signed-rank test) (Fig. 1A-B). In comparison, the genomic coverage of individual elements for these active histone modifications does not discriminate primitive from differentiated cells (Fig 1B). In contrast, H3K36me3, H3K27me3 and H3K9me3 derived LOCKs do not show any significant differences in the proportion of the genome covered between primitive and differentiated cells (FDR > 0.05; Wilcoxon signed-rank test) (Fig. 1A-B)."

The ES-derived were already mentioned and depending on which group the authors are considering them makes the variation more striking. It should be addressed in the text why some ES/iPS samples are lower than many differentiated cells. This could be a technical artifact of that ChIP-seq having lower enrichment, or something about the biology of that sample.

The ES-derived cells are considered as neither stem nor differentiated samples. They were not included in the analyses comparing primitive to differentiated cells. This is now clearly stated across the manuscript.

In terms of individual sample comparison between primitive and differentiated cells, while some ESC and iPSC samples have low H3K4/27ac coverage by LOCKs compared to the other ESC and iPSC samples, we kept them under the primitive group as previously assigned in the Roadmap Epigenomics Project manuscript (Kundaje et al 2015 Nature). This was done to

ensure direct comparison between our work and discoveries previously made when comparing the stem with the differentiated cells using the Roadmap Epigenomics Project and The ENCODE Project data.

REVIEWER COMMENTS

Reviewer #1 (Remarks to the Author):

In the revised version of the "Large Organized Chromatin Lysine Domains (LOCKS) Distinguish Primitive from Differentiated Cell Populations" the authors made some edits to the text and figures, though all the changes made I consider to be relatively minor.

A previous major concern I had was related to the novelty of the results and questioning the advantages or insights gained by conducting the analysis through LOCKS compared to more direct ways that other papers reporting similar results have previously used.

I think there is some value with the results presented in Fig 1. I think it is related to, but not exactly the same results that have been previously reported on differences between ES and more differentiated cell types that I am aware of. It also more directly makes use of the LOCK representation.

However, for the results of Fig. 2-5, while the authors are showing that previously reported results found by more direct means are also found with LOCKS, the authors haven't shown the results are specific to LOCKS. As LOCKS are defined from the underlying peaks it is not surprising that they can rediscover many results that can be found through peaks or other more direct means. There is no analysis showing the results don't also hold for signal outside of LOCKS.

Some comments on the additional comments:

*Fig. 4 only uses RNA-seq and ChIP-seq, and there was RNA-seq data available in many samples with the main Roadmap Epigenomics paper that provided the ChIP-seq data the authors are using

*The authors are still trying to contrast the active histone marks from the repressive histone marks by just being above or below a specific FDR threshold, this alone does not imply there is a significant difference between all the marks in the two sets.

Reviewer #2 (Remarks to the Author):

The authors have addressed all of my prior criticisms. I do suggest they further emphasize the distinctions between stem and differentiated and their categorization of the ES-derived. From a cell biology perspective, there are reasons to consider them not differentiated like a primary cell, but the grouping in each analysis needs to be especially clear so readers don't assume and misinterpret.

We appreciate the reviewers for their final concerns and comments to assure the high quality of the manuscript. We provided a description and implemented the necessary changes based on their comments.

Reviewer #1 (Remarks to the Author):

In the revised version of the “Large Organized Chromatin Lysine Domains (LOCKS) Distinguish Primitive from Differentiated Cell Populations” the authors made some edits to the text and figures, though all the changes made I consider to be relatively minor.

A previous major concern I had was related to the novelty of the results and questioning the advantages or insights gained by conducting the analysis through LOCKS compared to more direct ways that other papers reporting similar results have previously used.

I think there is some value with the results presented in Fig 1. I think it is related to, but not exactly the same results that have been previously reported on differences between ES and more differentiated cell types that I am aware of. It also more directly makes use of the LOCK representation.

However, for the results of Fig. 2-5, while the authors are showing that previously reported results found by more direct means are also found with LOCKS, the authors haven't shown the results are specific to LOCKS. As LOCKS are defined from the underlying peaks it is not surprising that they can rediscover many results that can be found through peaks or other more direct means. There is no analysis showing the results don't also hold for signal outside of LOCKS.

In Fig 3 and Fig 4C, the enrichment of pathways was conducted considering genes associated with peaks (individual elements) as the reference set. This is clarified in the methods section on page 16 of the revised manuscript:

“Hypergeometric test was used to identify p-values for enrichment of gene sets using dhyper function in stats R package (version 3.5.1). LOCK-associated genes in each sample are considered as query gene sets. In case of pathway enrichment per tissue type, catalogue of genes associated with all the individual elements were used as background gene lists. For the pathway enrichment across LOCKs with different H3K27me3 signal intensity in H1-hESC, all genes in proximity of LOCKs of the same histone mark in H1-hESC are considered as background gene lists.”

In Fig. 4A, expression of genes associated with LOCKS were compared directly with expression of genes associated with peaks (individual elements), as explained in the following statements in page 9 of the manuscript:

“Coexistence of active and repressive histone modifications at the same loci were reported in primitive cells as bivalent chromatin states associated with genes poised for expression or repression upon cellular differentiation 18. Hence, we assessed if bivalency is also related to LOCKs¹⁹. Overlapping repressive marks signal with LOCKs

from active and repressed chromatin across our collection of cell types revealed that bivalent LOCKs populate primitive cells, mapping in proximity to genes highly expressed, compared to genes in proximity of Ind. elements, only in differentiated cells, such as GM12878 and K562 as opposed to primitive H1-hESCs (Fig. 4A). ”

In Fig. 4B, the ChIP-seq signal were normalized to the ChIP-seq signal overlapped in the peaks (individual elements) as described initially in the caption of the figure as follows:

“Quantification of H3K27me3 ChIP-seq signal overlap in LOCKs of active histone modifications (H3K4me1, H3K4me3 and H3K27ac) for H1-hESC, GM12878 and K562 cell types. The signal is normalized (divided by median) to the H3K27me3 ChIP-seq signal overlapped in the Ind. elements of the corresponding profiles in each cell line.”

We have now revised the following paragraph in the methods section on page 16 to add this information:

“We identified overlap of signal from bedgraph files of ChIP-Seq data of H3K27me3 with the identified LOCKs and individual elements in GM12878, K562, and H1-hESC using bedtools (version 2.23.0). The signal over LOCKs was normalized (divided by median) to the H3K27me3 ChIP-seq signal overlapped in the Ind. elements of the corresponding profiles in each cell line. The identified signal intensity within each LOCK was then further normalized to the length of the element. The distribution of the normalized scores were then compared between H1-hESC and GM12878 and K562 cell lines using Wilcoxon signed-rank test.”

In Fig. 2, the clustering results based on LOCKs are meant to provide evidence for the high specificity of LOCKs in particular cell types with their sample phenotypes, not a new biological discovery to be compared with individual elements.

Some comments on the additional comments:

*Fig. 4 only uses RNA-seq and ChIP-seq, and there was RNA-seq data available in many samples with the main Roadmap Epigenomics paper that provided the ChIP-seq data the authors are using

Fig. 4 is focused on the tier I cell lines from The ENCODE Project.

*The authors are still trying to contrast the active histone marks from the repressive histone marks by just being above or below a specific FDR threshold, this alone does not imply there is a significant difference between all the marks in the two sets.

Significant differences reported in Fig 1A and B relate to the comparison between primitive and differentiated cells for each histone modification, independently of other histone modifications. To clarify our comparison between primitive and differentiated cells we have revised the manuscript to also report Fold change differences in the proportion of the genome covered by a particular histone modification between primitive and differentiated phenotypes in the following paragraph in page 4 of the manuscript:

“Comparing between cell types, LOCKs of the H3K4me1, H3K4me3 and H3K27ac active marks cover a larger proportion of the genome in primitive cells, including ESCs and iPSCs, compared to differentiated cells (non- ESCs and -iPSCs; FDR < 0.05; Wilcoxon signed-rank test; Fold change > 3.1) (Fig. 1A-B). In comparison, the genomic coverage of individual elements for these active histone modifications does not discriminate primitive from differentiated cells (Fig 1B). In contrast, H3K36me3, H3K27me3 and H3K9me3 derived LOCKs do not show any significant differences in the proportion of the genome covered between primitive and differentiated cells (FDR > 0.05; Wilcoxon signed-rank test; Fold change < 1.1) (Fig. 1A-B).”

Reviewer #2 (Remarks to the Author):

The authors have addressed all of my prior criticisms. I do suggest they further emphasize the distinctions between stem and differentiated and their categorization of the ES-derived. From a cell biology perspective, there are reasons to consider them not differentiated like a primary cell, but the grouping in each analysis needs to be especially clear so readers don't assume and misinterpret.

To further emphasize on our categorization of the samples, we revised the following statement in page 4 of the manuscript to include our definition of stem and differentiated samples:

“Comparing between cell types, LOCKS of the H3K4me1, H3K4me3 and H3K27ac active marks cover a larger proportion of the genome in primitive cells, including ESCs and iPSCs, compared to differentiated cells (non- ESCs and -iPSCs and -ES-derived; FDR < 0.05; Wilcoxon signed-rank test) (Fig. 1A-B).”

REVIEWERS' COMMENTS

Reviewer #1 (Remarks to the Author):

A main concern I previously raised was that the results of Fig. 2-5 are similar to what has been previously reported without using LOCKS and the authors were not showing the results were specific to LOCKS or clear advantages of this representation for these analyses.

In the authors response they acknowledged that for the results of Fig. 2 there is not a specific advantage for LOCKS. However, that is not acknowledged in the corresponding results section of Fig. 2 and there is no reference to the literature in that section of similar results being produced without LOCKS.

The author's response did not address this concern in the context of the results of Fig. 5.

For the results of Figs. 3 and 4C, the authors response makes the point that they used as a background only those genes which had an individual element assigned to it. That the authors were evaluating this conditional enrichment opposed to the actual enrichment of LOCKS was not stated in the results or figure legend, only in the methods section. That the LOCKS are significant after conditioning on genes that had an individual element, does not directly make the case for an advantage of LOCKS for identifying biological pathways, since one might still see greater overall significance for a similar set of categories without going through LOCKS. The LOCKS are likely identifying a more specific set of genes for some of these categories, but might also be missing more genes in them as well. Additionally, comparisons about LOCKS is confounded in that the individual elements are more restricted to being within the arbitrarily defined 10kb of gene, as LOCKS can more easily leverage signal outside this window given their broader definition. Furthermore, it is well established that promoter signal is less cell type specific than enhancer signal, so including any gene with any individual promoter element does not seem a particularly rigorous comparison for evaluating cell type specificity.

For the results of Figs. 4A and 4B, the authors make the point they are seeing greater expression or signal activity for LOCKS compared to individual elements. It is likely the authors are gaining specificity at the cost of sensitivity by using LOCKS in identifying biologically relevant elements. I expect one can also get a more specific set of individual elements by simply raising the threshold used to call them though it is possible those from LOCKS might be more specific at the same sensitivity.

Please find below our responses in bold to the reviewer's comments.

Review #1:

A main concern I previously raised was that the results of Fig. 2-5 are similar to what has been previously reported without using LOCKS and the authors were not showing the results were specific to LOCKS or clear advantages of this representation for these analyses.

In the authors response they acknowledged that for the results of Fig. 2 there is not a specific advantage for LOCKS. However, that is not acknowledged in the corresponding results section of Fig. 2 and there is no reference to the literature in that section of similar results being produced without LOCKS.

We added the following statement in the first paragraph of page 6 according to the suggestions:

“These results parallels previously reported improved stratification of primitive from differentiated cells using individual active compared to repressive elements⁶, suggesting that LOCKs are reflective of the underlying biology from individual elements.”

The author's response did not address this concern in the context of the results of Fig. 5.

Our results looking at LOCKs enrichment at TAD boundaries in Figure 5 parallels previous reports showing the preferential presence of cluster of cis-regulatory elements (COREs) over individual cis-regulatory elements at TADs (Madani Tonekaboni et al. 2019) and the preferential clustering of transcription factor binding sites, known as homotypic clusters of transcription regulator binding regions (HCTs), compared to individual binding sites at TAD boundaries (Madani Tonekaboni et al. 2019; Kentepozidou et al. 2020). We revised our manuscript to discuss our findings in light of previous work on page 9 as follows:

“Previous reports demonstrated the propensity of clusters of cis-regulatory elements and homotypic clusters of transcription factor binding regions (HCTs) to map at TAD boundaries (Madani Tonekaboni et al. 2019; Kentepozidou et al. 2020).”

We also clarify that our analysis focused on LOCKs and TADs, not in comparison to individual elements on page 9 as follows:

“We therefore investigated the relation between LOCKs, independently from individual elements, and the three-dimensional genome organization of primitive versus differentiated cells”

For the results of Figs. 3 and 4C, the authors response makes the point that they used as a background only those genes which had an individual element assigned to it. That the authors were evaluating this conditional enrichment opposed to the actual enrichment of LOCKS was not stated in the results or figure legend, only in the methods section.

We actually used the catalogue of genes associated with all individual elements as background for our analysis. We have now addressed this comment by further clarifying how we performed the analysis with the following statement to the legend of Figure 3:

“In this analysis, the catalogue of genes associated with all the individual elements were used as background gene lists.”

and to Figure 4C:

“In this analysis, the catalogue of genes associated with all the individual elements, with corresponding H3K27me3 signal level, were used as background gene lists.”

That the LOCKS are significant after conditioning on genes that had an individual element, does not directly make the case for an advantage of LOCKS for identifying biological pathways, since one might still see greater overall significance for a similar set of categories without going through LOCKS.

The LOCKS are likely identifying a more specific set of genes for some of these categories, but might also be missing more genes in them as well. Additionally, comparisons about LOCKS is confounded in that the individual elements are more restricted to being within the arbitrarily defined 10kb of gene, as LOCKS can more easily leverage signal outside this window given their broader definition.

Our results do not exclude the value of individual elements but highlights the benefit of taking into account LOCKs, as shown throughout the manuscript and in direct comparison to individual elements in Figure 4. This is further addressed with the updates in the manuscript presented in the response to the first comment.

Furthermore, it is well established that promoter signal is less cell type specific than enhancer signal, so including any gene with any individual promoter element does not seem a particularly rigorous comparison for evaluating cell type specificity.

We processed each histone modification independently of each other to assess the utility of LOCKS to discriminate subtypes. These include some histone modifications whose distributions are skewed to promoters and other skewed away from promoters. Our results argue that LOCKS from active marks, independently of whether their distribution is skewed to promoters or not, stratify primitive from differentiated cell types and cluster each sample according to their tissue of origin.

For the results of Figs. 4A and 4B, the authors make the point they are seeing greater expression or signal activity for LOCKS compared to individual elements. It is likely the authors are gaining specificity at the cost of sensitivity by using LOCKS in identifying biologically relevant elements. I expect one can also get a more specific set of individual elements by simply raising the threshold used to call them though it is possible those from LOCKS might be more specific at the same sensitivity.

In contrast to other tools looking at clusters of individual elements, such as ROSE used to call Super-enhancers, our approach is based on identifying clusters of peaks independently of the ChIP-seq signal intensity. While lowering peak detection threshold or restricting the analysis to the top 500 peaks may increase the specificity of the data, identifying LOCKS within these higher-stringency peaks will simply result in fewer LOCKS being identified.